# Genetic Reassortment between Endemic and Introduced *Macrobrachium rosenbergii* Nodaviruses in the Murray-Darling Basin, Australia

**DOI:** 10.3390/v14102186

**Published:** 2022-10-04

**Authors:** Vincenzo A. Costa, Jemma L. Geoghegan, Edward C. Holmes, Erin Harvey

**Affiliations:** 1Sydney Institute for Infectious Diseases, School of Medical Sciences, The University of Sydney, Sydney, NSW 2006, Australia; 2Department of Microbiology and Immunology, University of Otago, Dunedin 9016, New Zealand; 3Institute of Environmental Science and Research, Wellington 5022, New Zealand

**Keywords:** metagenomics, virus discovery, *Nodaviridae*, *Macrobrachium rosenbergii* nodavirus, virus reassortment

## Abstract

*Macrobrachium rosenbergii* nodavirus (MrNV)—the aetiological agent of white tail disease—is a major limiting factor of crustacean aquaculture as it causes up to 100% mortality in *M. rosenbergii* larvae and juveniles. Despite the importance of MrNV, there have been few studies on the phylogenetic diversity and geographic range of this virus in Australian waterways. Here, we detected MrNV genomes in common carp (*Cyprinus carpio*) metatranscriptomes sampled at five freshwater sites across the Murray-Darling Basin (MDB), Australia. We identified genetic divergence of the RNA-dependent RNA polymerase gene between MrNV sequences identified in the northern and southern rivers of the MDB. Northern viruses exhibited strong phylogenetic clustering with MrNV from China, whereas the southern viruses were more closely related to MrNV from Australia. However, all five viruses were closely related in the capsid protein, indicative of genetic reassortment of the RNA1 and RNA2 segments between Australian and introduced MrNV. In addition, we identified *Macrobrachium australiense* in two of the five MrNV-positive libraries, suggesting that these species may be important reservoir hosts in the MDB. Overall, this study reports the first occurrence of MrNV outside of the Queensland region in Australia and provides evidence for genetic reassortment between endemic and introduced MrNV.

## 1. Introduction

Decapod crustaceans (crabs, shrimps, and prawns) form an important component of the human diet, contributing approximately 11.2 million tonnes to global aquaculture production [1]. Shrimps and prawns are among the most internationally traded aquatic species, with exported products valued at 24.7 billion US dollars in 2020 [1]. The transboundary movement of these species is a key ecological driver of emerging infectious diseases [2]. Viruses are a major limiting factor on crustacean aquaculture, responsible for substantial economic losses of up to 3 billion US dollars annually [2]. The World Organisation for Animal Health (OIE) currently lists ten notifiable diseases of crustaceans, seven of which are caused by viruses including *Macrobrachium rosenbergii* nodavirus (MrNV) [3].

MrNV is an icosahedral nonenveloped positive-sense single-stranded RNA virus within the family *Nodaviridae* [4]. Its genome is arranged into two segments, RNA1 and RNA2. The RNA1 segment (approximately 3.2 kb) contains two open reading frames (ORFs) encoding RNA-dependent RNA polymerase and a subgenomic B2 protein that suppresses host antiviral immunity [5]. RNA2 is a smaller segment (approximately 1.2 kb) that encodes the capsid protein [5]. MrNV infection is often coupled with a positive-sense RNA satellite virus—extra small virus (XSV; *Sarthroviridae*)—of approximately 796 nucleotides encoding two capsid proteins, CP-17 and CP-16 [6].

Both MrNV and XSV cause white tail disease (WTD) in various freshwater and marine decapods, including *Penaeus* species, *Cherax quadricarinatus*, *M. malcolmsonii*, *M. rude*, and most notably, *M. rosenbergii* [7,8,9]. WTD primarily impacts *M. rosenbergii* larvae, post-larvae and juveniles causing a mortality rate of up to 100% in hatcheries and farms [5]. MrNV can be transmitted horizontally in the aquatic environment where it may be occasionally vectored by various invertebrate species including *Artemia* brine shrimps [10].

WTD was first reported from a hatchery located in the French West Indies during 1999, then later identified in India, China, Taiwan, Malaysia, Thailand, Indonesia, and Australia [11,12,13,14,15,16,17,18]. Despite the importance of WTD in the Australasian region, there have been few studies on the genetic diversity and geographic range of this virus in Australian waterways. Indeed, there remains only one Australian MrNV genome available, identified in Queensland [17].

Here, we describe five novel MrNV genomes and one genome of XSV identified in the metatranscriptomes of common carp (*C. carpio*) sampled across northern and southern rivers of the Murray-Darling Basin (MDB), Australia [19]. Using these data, we aim to provide a better understanding on the phylogenetic range and distribution of MrNV in Australian waterways.

## 2. Materials and Methods

### 2.1. MrNV Discovery and Genome Annotation

Metatranscriptomes (n = 13) of the common carp (*C. carpio*) were screened for the presence of MrNV. These transcriptomes were generated as part of a large survey of fish viruses in the MDB during January and March 2020 (see [19] for details of RNA extraction and sequencing). These reads are available on the NCBI Sequence Read Archive under BioProject PRJNA701716.

Sequencing reads were quality trimmed using Trimmomatic v.0.39 [20] and de novo assembled into contigs using Trinity RNA-seq v.2.8.5 [21]. The contigs generated were used as a query against the NCBI nucleotide (nt) and non-redundant (nr) databases using BLASTn and Diamond (BLASTx) [22]. Contigs with BLAST hits (i.e., 95–100% sequence identity) to MrNV were predicted into ORFs using Geneious Prime (v.2022.0) [23]. MrNV predicted ORFs were then used as a second query against the NCBI nt and nr databases using BLAST implemented in Geneious for validation. MrNV genomes were annotated using the NCBI conserved domain (CDD) search and the ‘Live Annotate and Predict’ function in Geneious Prime using all published MrNV genomes available on NCBI/GenBank as reference sequences. Amino acid sequences of the RNA1 and RNA2 segments were further annotated using InterProScan [24]. All virus genomes generated have been deposited in NCBI/GenBank under the accessions OP508292–OP508302.

### 2.2. Estimation of Transcript Abundance

Transcript abundance was calculated using RNA-seq by Expectation-Maximization (RSEM) (v1.2.28) [21]. To determine the relative abundance of MrNV transcripts in the metagenomic dataset, abundances were standardized by the total number of paired reads for each given library.

### 2.3. Phylogenetic Analysis

To determine the genetic diversity of MrNV sequences within the MDB and globally, we estimated phylogenetic trees using nucleotide sequences of RNA-dependent RNA polymerase (RdRp) and the capsid protein. We aligned our sequences with all MrNV genomes available on NCBI/GenBank using MAAFT v.7.450 implemented in Geneious with the E-INS-i algorithm [25]. Sequence alignments were pruned using trimAl (v.1.2) [26]. The best-fit model of nucleotide substitution was determined using the ‘ModelFinder Plus’ function (-m MFP) implemented in IQ-TREE v.1.6.8 [27,28]. Maximum likelihood trees were estimated using IQ-TREE with 1000 bootstrap replicates [27].

### 2.4. Screening for Invertebrate Sequences in Fish Metagenomes

To identify any possible contaminant sequences—that may represent potential decapod hosts or aquatic insect vectors [10]—we aligned our contigs to a custom database containing all nucleotide sequences available on NCBI (except for environmental or artificial sequences) using the KMA aligner and CCMetagen [29,30]. Sequences with a gene identity of less than 95% to arthropod species were omitted from the analysis. Abundance measures and Krona graphs were generated using CCMetagen [30].

## 3. Results

### 3.1. Composition of Metatranscriptomes and Identification of Macrobrachium australiense

Metagenomic classification of total reads (range 67,004,988–95,168,951) in each transcriptome revealed that *C. carpio* accounted for 97–99% of total reads (Figure 1). This analysis also identified the presence of *M. australiense* at the Macquarie River and Murray River (Wemen) sites (Figure 1 and Figure 2). *M. australiense* sequences made up 0.01% of the reads in the Macquarie River library and 0.003% in the Murray River (Wemen) library. The reads identified had 99–100% identity to *M. australiense* (NCBI/GenBank: AY374145.1).

### 3.2. Presence of MrNV and XSV

BLAST identified contigs representing the genomes of five novel MrNV variants in common carp (*C. carpio*) metatranscriptomes [19]. All five genomes were identified at different freshwater sites—two north and three south—across the MDB (Figure 2; Table 1). MrNV sequences exhibited an average standardised abundance of 0.01% (range 0.002–0.07%) (Table 1). In addition, we detected the presence of the satellite virus, XSV, at the Murray River (Wemen) site.

### 3.3. Genome Organization

All five MrNV genomes exhibited the typical nodavirus RNA1 and RNA2 segments. RNA1 segment contained a non-structural ORF encoding RdRp (1038 amino acids), and a subgenomic RNA containing the B2 protein (133 amino acids) (Figure 3). The RdRp ORF contained a conserved methyltransferase domain, likely involved in mRNA capping [31] and an RdRp domain (Figure 3). The RNA2 segment contained the capsid protein, with a length of 371 amino acids. In addition, XSV exhibited a genome of 823 nt with two overlapping structural proteins, CP-17 (175 amino acids) and CP-16 (95 amino acids) (Figure 3).

### 3.4. MrNV Nucleotide Similarity between Northern and Southern Rivers of the MDB

All five genomes identified in this study exhibited 95–99% RdRp and 92–99% capsid nucleotide similarity to all currently described *M. rosenbergii* nodaviruses (Table 1). The sequences of the northern rivers—Barwon and Macquarie—exhibited 99.2% RdRp and 99.8% capsid nucleotide identity. Similarly, all three sequences of the southern rivers—Edward River and Murray River sites—exhibited high RdRp (99.2%) and capsid (99.6%) similarity. Notably, we detected higher levels of genetic diversity across the RdRp gene between nothern and southern sequences (95.9% similarity) compared to the capsid (99.7%) (Appendix A).

### 3.5. Phylogenetic Relationships

Phylogenetic analysis of the RdRp gene revealed two MrNV clades, one containing viruses from China, India, Malaysia, and the French West Indies, and the other, viruses from Australia (Figure 4). Both northern MDB sequences (Barwon and Macquarie rivers) cluster within the former clade, exhibiting 99% nucleotide similarity with MrNV from China (NCBI/GenBank: FJ751226.1). Conversely, sequences from the southern region (Edward and Murray rivers) exhibit strong phylogenetic clustering with Australian MrNV (NCBI/GenBank: JN619369.1).

Similarly, the phylogeny of the capsid gene revealed two clades, with genetic divergence between the Australian, Indian, Chinese, and French West Indies viruses (Figure 5). However, all five MDB sequences fall within the Australian clade, sharing 98–99% nucleotide similarity with MrNV from Australia (NCBI/GenBank: JN619370.1) and with strong (95%) bootstrap support. Such phylogenetic incongruence is indicative of genetic reassortment among the RNA1 and RNA2 segments.

## 4. Discussion

Investigation of *C. carpio* metatranscriptomes enabled us to identify five novel MrNV genomes and one XSV genome in the MDB [19]. This is the first time this virus has been detected outside of the Queensland region in Australia where it causes white tail disease in the giant freshwater prawn (*M. rosenbergii*) [17,18]. All five genomes exhibited typical MrNV genomic organization containing the RNA1 and RNA2 segments encoding RdRp, B2 and the capsid protein (Figure 3).

While we detected MrNV in combined liver and gill samples of the common carp (*C. carpio*), it is likely that these viruses were derived from environmental contamination of gill tissue, which often leads to the identification of a large diversity of viruses associated with aquatic invertebrates [32,33]. This is supported by the identification of *M. australiense* in two of the five MrNV-positive libraries (Figure 1). *M. australiense* is an endemic freshwater prawn and is one of the most widely distributed invertebrate species in Australian freshwater ecosystems, occupying most of the eastern and north-western waterways [34]. Given that MrNV causes asymptomatic infections in other *Macrobrachium* species—*M. malcolmsonii*, and *M. rude*—these results suggest that *M*. *australiense* may be an important reservoir host in the MDB, particularly as we detected this association across a large geographic area (i.e., separated by approximately 760 kilometres) (Table 1; Figure 2) [8].

Notably, we identified genetic divergence of the RdRp gene between MrNV detected in the northern and southern rivers of the MDB (Figure 4). That the northern RdRp sequences clustered with those from China, India, and the French West Indies, strongly suggests these viruses were introduced into the MDB through the international trade. Indeed, shrimps and prawns are the second highest traded aquatic animal group globally, accounting for 16% of all exported species (salmons and trouts being the highest at 18%) [1]. Furthermore, it is important to note that all five capsid sequences formed a distinct and strongly supported clade with Australian MrNV, sister to the viruses obtained from all other countries (Figure 5). This is indicative of genetic reassortment of the RNA1 and RNA2 segments between endemic and introduced viruses in the MDB.

Viral reassortment—a mechanism of genetic recombination in segmented RNA viruses—can lead to novel viral genomes through the exchange of gene segments during co-infection [35]. All families of segmented RNA viruses are capable of generating reassortant viruses, some of which are often associated with cross-species transmission and disease emergence (e.g., influenza viruses) [36]. Given their bipartite segmented genomes, genetic reassortment of the RNA1 and RNA2 segments has also been observed in the genus *Betanodavirus* [37]. For example, reassortment (RSGNNV/SJNNV) between the red-spotted grouper nervous necrosis virus (RSGNNV) and striped jack nervous necrosis virus (SJNNV) is an emerging pathogen in farmed fish [38].

In sum, this study reports the first occurrence of MrNV in the MDB, the largest freshwater river system in Australia. While we cannot determine whether the viruses identified in this study were associated with WTD, it is important to note that we detected the presence of both MrNV and XSV at the Murray River (Wemen) site, indicative of co-infection [39] (Figure 2 and Figure 3). Furthermore, the identification of such viruses across a large geographic area suggests that freshwater prawns inhabiting the MDB (e.g., *M. australiense*) may be important reservoir hosts for MrNV. Further studies are clearly required to determine host susceptibility as well as the pathogenicity and transmissibility of the reassortant viruses identified here.

## Figures and Tables

**Figure 1 viruses-14-02186-f001:**
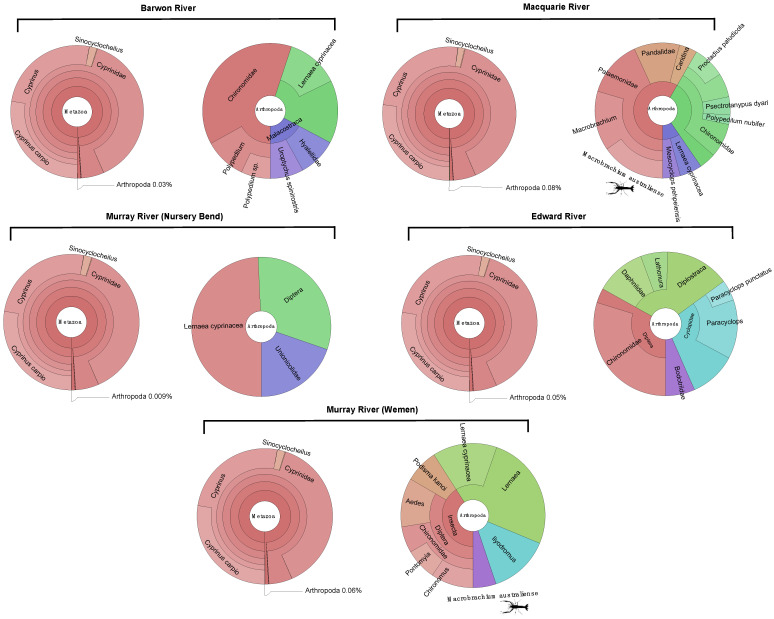
Krona plots showing the relative abundances of *C. carpio* and arthropod reads in MrNV positive libraries. Each library contains one Krona plot broadly showing the abundance of arthropod compared to fish reads, and another showing the distribution of all arthropod reads. Each library is labelled with its corresponding site.

**Figure 2 viruses-14-02186-f002:**
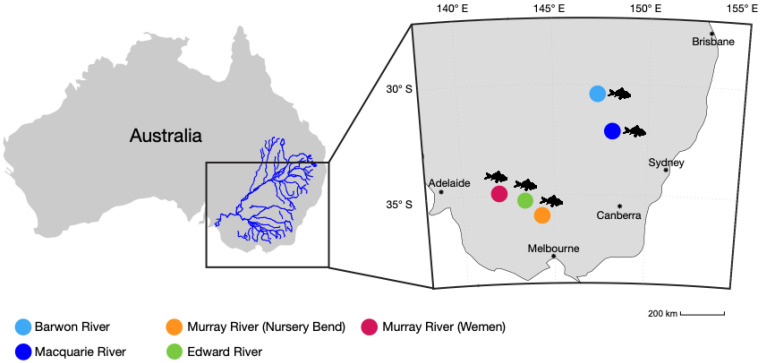
Locations across the MDB where all MrNV genomes were discovered. Coloured circles correspond to the site name and fish silhouettes represent *C. carpio*.

**Figure 3 viruses-14-02186-f003:**
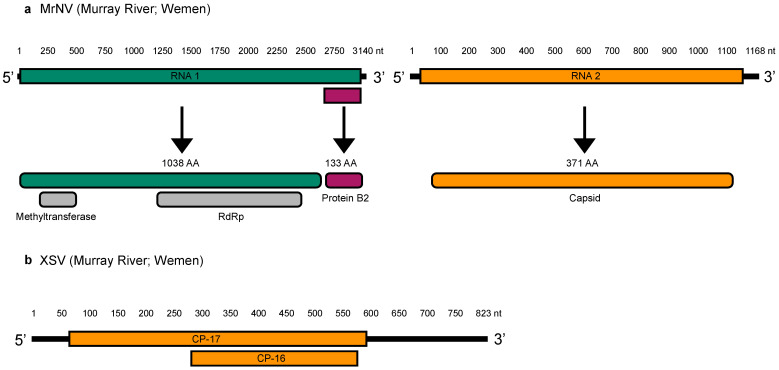
(**a**) Genome organization of novel MrNV detected at the Murray River (Wemen) site. Top bars represent RNA1 and RNA2 segments, with arrows showing encoded proteins and amino acid length. Grey bars represent conserved domains along with RdRp ORF. (**b**) Genome organization of XSV. Orange bars illustrate overlapping genes of structural proteins CP-17 and CP-16. The MrNV genome from the Murray River (Wemen) site was selected for illustration purposes as this was the only library that also contained the satellite virus, XSV.

**Figure 4 viruses-14-02186-f004:**
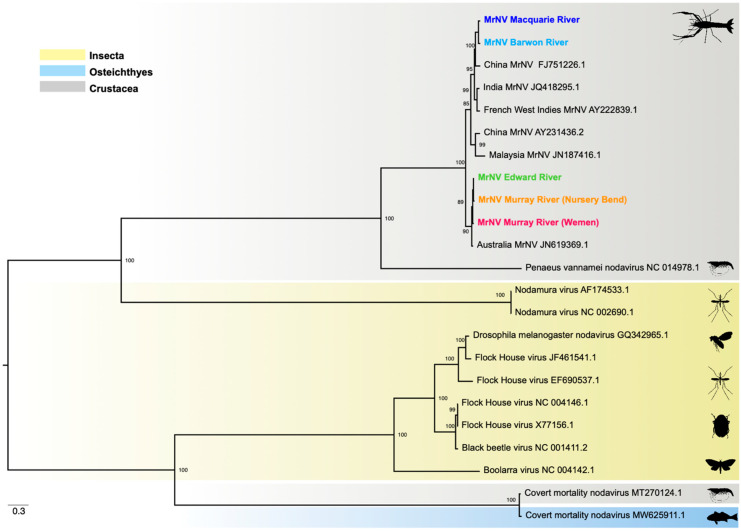
Maximum likelihood tree of the RdRp gene (3126 bp) of MrNV and related alphanodavirus sequences. The viruses detected are colour-coded to match Figure 1. The phylogeny is midpoint- rooted for clarity only and scale bar represents nucleotide substitutions per site. Tip labels represent virus name and NCBI/GenBank accession ID.

**Figure 5 viruses-14-02186-f005:**
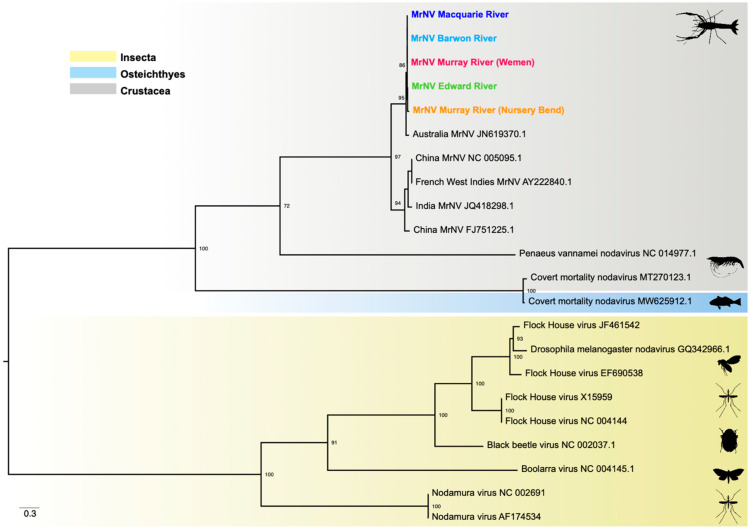
Maximum likelihood tree of the capsid gene (1116 bp) of MrNV and related alphanodavirus sequences. Detected viruses are colour-coded to match Figure 1. The phylogeny is midpoint-rooted for clarity only and scale bar represents nucleotide substitutions per site. Tip labels represent virus name and NCBI/GenBank accession ID.

**Table 1 viruses-14-02186-t001:** Description of novel MrNV and XSV genomes identified in the MDB.

Virus	Site/NCBI SRA Accession	Coordinates	Gene	Length (nt)	Standardised Abundance (%)	Closest Relative (GenBank)	Identity (%)
MrNV	Barwon River/SRX10098043	29°58′26.8″ S 148°06′45.5″ E	RdRp	3126	0.052	China MrNV (FJ751226.1)	98.1
			Capsid	1116	0.022	Australia MrNV (JN619370)	98.8
MrNV	Macquarie River/SRX10098050	31°59′10.7″ S 148°12′37.8″ E	RdRp	3126	0.003	China MrNV (FJ751226.1)	98
			Capsid	1116	0.002	Australia MrNV (JN619370)	98.8
MrNV	Murray River (Nursery Bend)/SRX10098058	35°45′58.0″ S 144°20′55.0″ E	RdRp	3126	0.001	Australia MrNV (JN619369)	99.1
			Capsid	1116	0.001	Australia MrNV (JN619370)	98.5
MrNV	Murray River (Wemen)/ SRX10098067	34°45′55.6″ S 142°38′44.0″ E	RdRp	3126	0.002	Australia MrNV (JN619369)	99.3
			Capsid	1116	0.001	Australia MrNV (JN619370)	98.9
XSV			Capsid	525	6.3 × 10^−5^	Thailand XSV (NC_043498)	99
MrNV	Murray River (Edward River)/SRX10098059	35°05′28.8″ S 144°00′36.4″ E	RdRp	3126	0.009	Australia MrNV (JN619369)	99.2
			Capsid	1116	0.004	Australia MrNV (JN619370)	98.7

## Data Availability

All MrNV and XSV genomes have been deposited in NCBI/GenBank under the accessions OP508292–OP508302.

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
