# Peer review of "Genetic Reassortment between Endemic and Introduced Macrobrachium rosenbergii Nodaviruses in the Murray-Darling Basin, Australia"

_viruses, 2022, doi:10.3390/v14102186_

Round 1
Reviewer 1 Report
The MS entitled " Genetic Reassortment Between Endemic and Introduced Mac- 2 robrachium rosenbergii Nodaviruses in the Murray-Darling Ba- 3 sin, Australia" is very interesting research. The MS is well planned and organized and suitable for publication
Author Response
We thank the reviewer for their comment on the manuscript.
Reviewer 2 Report
In the present study, the authors detected MrNV genomes in common carp (Cyprinus carpio) metatranscriptomes sampled at five freshwater sites across the Murray-Darling Basin (MDB), Australia. The genetic divergence of the RNA-dependent RNA polymerase gene between MrNV sequences identified in the northern and southern rivers of the MDB was also identified.The topic is very interesting, but needs to add some data.
1. Further testing of the identified virus is needed.
2. Line 82, the accession numbers should be added.
3. Line 197-199, That the northern RdRp sequences clustered with those from China, India and the French West Indies, strongly suggests these viruses were introduced into the MDB through the international trade. The international trade is an aspect, whether the environment will also affect?
Author Response
We thank the reviewer for their comments. We will address their three points below:
- Further testing of the identified virus is needed.
It is unclear what exactly 'further testing' the reviewer feels is necessary, or what they would like us to clarify through this further testing, so we are unable to address this comment within the timeframe provided for our revisions. Indeed, it is impossible to perform PCR confirmation experiments within this time frame. - Line 82, the accession numbers should be added.
We have now added the accession numbers to the revised manuscript as requested. - Line 197-199, That the northern RdRp sequences clustered with those from China, India and the French West Indies, strongly suggests these viruses were introduced into the MDB through the international trade. The international trade is an aspect, whether the environment will also affect?
Although our method does not conclusively determine that reassortment has taken place, we do believe that it is entirely reasonable for us to suggest this to explain the observation that the RdRp and capsid sequences cluster in different phylogenetic positions. We strongly feel that this is the most plausible explanation. It is unclear what environmental effect the reviewer believes could be responsible for this observation. We would therefore like to retain the original text.